# An Empirical Study of Graph Contrastive Learning

**Yanqiao Zhu**[1,2]    **Yichen Xu**[3]    **Qiang Liu**[1,2]    **Shu Wu**[1,2]

[1]Center for Research on Intelligent Perception and Computing
Institute of Automation, Chinese Academy of Sciences
[2]School of Artificial Intelligence, University of Chinese Academy of Sciences
[3]School of Computer Science, Beijing University of Posts and Telecommunications
yanqiao.zhu@cripac.ia.ac.cn, linyxus@bupt.edu.cn
{qiang.liu, shu.wu}@nlpr.ia.ac.cn

## Abstract

Graph Contrastive Learning (GCL) establishes a new paradigm for learning graph representations without human annotations. Although remarkable progress has been witnessed recently, the success behind GCL is still left somewhat mysterious. In this work, we first identify several critical design considerations within a general GCL paradigm, including augmentation functions, contrasting modes, contrastive objectives, and negative mining techniques. Then, to understand the interplay of different GCL components, we conduct extensive, controlled experiments over a set of benchmark tasks on datasets across various domains. Our empirical studies suggest a set of general receipts for effective GCL, e.g., simple topology augmentation that produces sparse graphs brings most performance improvements; contrasting modes should be aligned with the granularities of end tasks. In addition, to foster future research and ease the implementation of GCL algorithms, we develop an easy-to-use toolbox PyGCL, featuring modularized CL components, standardized evaluation, and experiment management. We envision this work to provide useful empirical evidence of existing GCL architectures and offer several insights for future research.

## 1   Introduction

The past few years have seen rapid advances in Graph Neural Networks (GNNs) [1, 2], which is a powerful framework for analyzing graph-structured data. As the most GNN models focus on (semi-)supervised learning, which requires access to abundant labels, recent trends in Self-Supervised Learning (SSL) have led to a proliferation of studies that learns graph representations without relying on human annotations. Among SSL methods, Contrastive Learning (CL), also known as instance discrimination, is a major area of interest and has already achieved comparable performance with its supervised counterparts in many representation learning tasks [3–12].

Recently, remarkable progress has been made to adapt CL techniques for the graph domain. A typical Graph Contrastive Learning (GCL) framework firstly constructs multiple graph views via stochastic augmentation of the input, and then learns representations by contrasting positive samples against negative ones. For each node being an anchor instance, positive samples are often chosen as its congruent representations in other views, while negatives are selected from other nodes within the given graph in node-centric datasets or other graphs in graph-oriented datasets. Although GCL has constituted a new paradigm of SSL in the graph domain and achieved promising results, recent studies [13–19] seem to resemble each other with very limited nuances from the methodological perspective. Moreover, most existing work only provides model-level evaluation. Still, the contributing factors leading to the success of GCL remain somewhat mysterious, which calls for a deeper understanding of different GCL components.

Preprint. Under review.

Towards this end, we try to shed light on the success behind these GCL algorithms through the lens of empirical evaluation of critical design considerations in existing work. We first propose a general contrastive paradigm and characterize previous work by limiting the design space of interest to four dimensions: (a) data augmentation functions, (b) contrasting modes, (c) contrastive objectives, and (d) negative mining strategies. Note that we include no model-specific design considerations such as the number of attention heads for graph attentive encoders. To the best that we are aware of, these four dimensions cover a wide range of options that are representative in a large body of literature.

Then, we systematically study the empirical performance of different design dimensions through controlled experiments over three benchmark tasks on a set of datasets across a variety of domains. Our empirical studies attempt to provide answers to the following questions:

- What is the most contributory component in an effective graph CL algorithm?
- How do different design considerations affect the model performance?
- Do these design considerations favor certain types of data or end tasks?

We note that although there has been several survey papers on self-supervised graph representation learning [20–22], to the best of our knowledge, none of existing work provides rigorous empirical evidence on the impact of each component in GCL.

We summarize several key findings of the empirical study, which we hope could benefit the graph SSL community for developing future algorithms. Our experiments suggest a set of general recipes for effective GCL algorithms:

- GCL algorithms benefit most from topology augmentations that produce sparse graph structures. In addition, bi-level augmentation on both topology and feature levels further improves the performance, if informative node attributes are available.
- The contrasting modes should be chosen according to the granularity of downstream tasks.
- The InfoNCE objective achieve stable, consistent performance improvements under all settings yet requires a large number of negative samples.
- Several recently proposed negative-sample-free objectives have great potential in reducing computational burden, while obtaining promising performance.
- Current negative mining strategies bring limited performance improvements to GCL.

In addition, to foster future research, we develop PyGCL, an easy-to-use PyTorch framework, featuring commonly used, modularized GCL components, standardized evaluation, and experiment management. We hope the use of PyGCL will greatly relief the burden of comparing existing baselines and developing new algorithms. The PyGCL is open-sourced at https://github.com/GraphCL/PyGCL.

## 2 A General Paradigm of GCL and its Design Dimensions

**Problem formulation.** Let $\mathcal{G} = (\mathcal{V}, \mathcal{E})$ denote a given graph, where $\mathcal{V} = \{v_i\}_{i=1}^N$, $\mathcal{E} \subseteq \mathcal{V} \times \mathcal{V}$ represent the node set and the edge set respectively. We further denote the feature matrix and the adjacency matrix as $\boldsymbol{X} \in \mathbb{R}^{N \times F}$ and $\boldsymbol{A} \in \{0, 1\}^{N \times N}$, where $\boldsymbol{x}_i \in \mathbb{R}^F$ is the feature of $v_i$, and $\boldsymbol{A}_{ij} = 1$ iff $(v_i, v_j) \in \mathcal{E}$. In the setting of unsupervised graph representation learning, there is no given class information of nodes in $\mathcal{G}$ during training. Our objective is to learn a GNN encoder $f$ receiving the graph features and structure as input, that produces node embeddings in low dimensionality. We denote $\boldsymbol{H} = f(\boldsymbol{X}, \boldsymbol{A}) \in \mathbb{R}^{N \times F'}$ as the learned representations of nodes, where $\boldsymbol{h}_i$ is the embedding of node $v_i$. For graph-oriented tasks, we can further obtain a graph-level representation $\boldsymbol{s} = r(\boldsymbol{H}) \in \mathbb{R}^{F'}$ of $\mathcal{G}$ that aggregates node-level embeddings. Note that the readout function $r$ might be a simple permutation-invariant function such as mean or sum pooling, or may be learnable and parameterized by a neural network. These representations can be used in downstream tasks, such as node/graph classification and community detection.

**General paradigm of GCL.** We decompose representative GCL algorithms from four dimensions: (a) data augmentation functions, (b) contrastive mode, (c) contrastive objective, and (d) negative mining strategies. These four components constitute the design space of interest for GCL in this work.

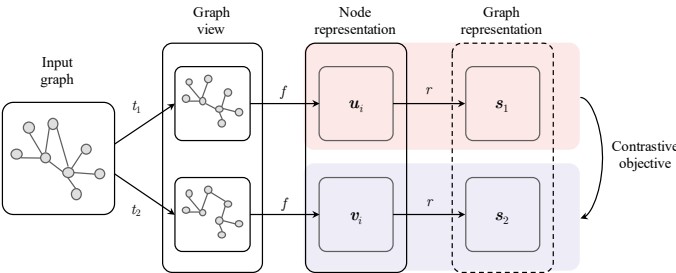

Figure 1: A general GCL model. At first, two graph views are generated via data augmentation functions. Then, the two graphs are fed into a shared graph neural network to learn representations, which are then optimized with a contrastive objective that pulls congruent representation pairs in the two views together while pushing others away. Additional negative sampling techniques may be employed to improve the model performance.

At each iteration of training, we first perform stochastic **augmentation** to generate multiple graph views from the input graph. Specifically, we sample two augmentation functions $t_1, t_2 \sim \mathcal{T}$ to generate graph views $\widetilde{\mathcal{G}}_1 = t_1(\mathcal{G})$ and $\widetilde{\mathcal{G}}_2 = t_2(\mathcal{G})$, where $\mathcal{T}$ is the set of all possible transformation functions to be discussed in the next section. We then obtain node representations for the two views using the shared GNN encoder $f$, denoted by $\boldsymbol{U} = f(\widetilde{\boldsymbol{X}}_1, \widetilde{\boldsymbol{A}}_1)$ and $\boldsymbol{V} = f(\widetilde{\boldsymbol{X}}_2, \widetilde{\boldsymbol{A}}_2)$ respectively. Optionally, we may obtain graph representations for each graph view: $\boldsymbol{s}_1 = r(\boldsymbol{U})$ and $\boldsymbol{s}_2 = r(\boldsymbol{V})$.

For every node embedding $\boldsymbol{v}_i$ being the anchor instance, the **contrasting mode** specifies a positive set $\mathcal{P}(\boldsymbol{v}_i) = \{\boldsymbol{p}_i\}_{i=1}^{P}$ and a negative set $\mathcal{Q}(\boldsymbol{v}_i) = \{\boldsymbol{q}_i\}_{i=1}^{Q}$. In a pure unsupervised learning setting, we only consider congruent samples in each graph view; in other words, embeddings in the two augmented graph views corresponding to the same node or graph constitute the positive set. Thus, we denote the only positive sample as $P(\boldsymbol{v}_i)$ afterwards for simplicity. It is noted that when label supervision is given, the positive set may be enlarged with samples belonging to the same class [23]. Moreover, we may employ **negative mining strategies** to improve the negative sample set by considering the relative similarity (i.e. the hardness) of negative samples. Finally, we use a **contrastive objective** $\mathcal{J}$ to score these specified positive and negative pairs and train the model in an unsupervised manner.

In the following we succinctly list implementations of these four design dimensions considered in this work. For details of each implementation, we refer readers of interest to Appendix E.

## 2.1 Design Dimensions

**Data augmentation.** The purpose of data augmentation is to generate *congruent, identity-preserving positive samples* of the given graph. Most graph CL work involves bi-level augmentation techniques: topology (structure) transformation and feature transformation. In this work we consider eight topology augmentation functions and two feature augmentations.

- **Topology augmentation:** (1) Edge Removing (ER), (2) Edge Adding (EA), (3) Edge Flipping (EF), (4) Node Dropping (ND), (5) Ego-net Sampling (ES), (6) Subgraph induced by Random Walks (RWS), (7) diffusion with Personalized PageRank (PPR), and (8) diffusion with Markov Diffusion Kernels (MDK).
- **Feature augmentation:** (1) Feature Masking (FM) and (2) Feature Dropout (FD).

**Contrasting modes.** For an anchor instance, contrasting modes determine the positive and negative sets at different granularities of the graph. In mainstream work, three contrasting modes are widely employed: (1) local-local CL and (2) global-global CL, which contrast embeddings at the same scale, and (3) global-local CL, which contrasts cross-scale embeddings.

**Contrastive objectives.** Contrastive objectives are used to train the encoder to maximize the agreement between positive samples and the discrepancy between negatives. We consider the following objective functions in this work: (1) InfoNCE, (2) Jason-Shannon Divergence (JSD), and (3) Triplet Margin loss (TM).

Table 1: Summary of representative graph CL models within the proposed paradigm.

| Method | Primary task | Topology augmentation | Feature augmentation | Contrasting mode | Dual branches? | Contrastive objective |
|---|---|---|---|---|---|---|
| DGI [13] | Node classification | ES | — | Global-local | ✗ | JSD |
| GMI [15] | Node classification | ES | — | Global-local | ✗ | SP-JSD |
| InfoGraph [19] | Graph classification | ES | — | Global-local | ✗ | SP-JSD |
| MVGRL [14] | Node & graph classification | PPR | — | Global-local | ✓ | JSD |
| GCC [24] | Transfer learning | RWS + ES | — | Local-local | ✗ | InfoNCE |
| GraphCL [16] | Node & graph classification | RWS/ND/EA/ED | FD | Local-local | ✓ | InfoNCE |
| GRACE [17] | Node classification | ER | MF | Local-local | ✓ | InfoNCE |
| GCA [18] | Node classification | ER | MF | Local-local | ✓ | InfoNCE |
| BGRL [25] | Node classification | ER | MF | Local-local | ✓ | BL |
| GBT [26] | Node classification | ER | MF | Local-local | ✓ | BT |

**Negative mining strategies.** Most existing work presumes embeddings of nodes or graphs other than the anchor instance to be dissimilar to the anchor and thus considers them as negatives. In this work, we consider the following four negative mining schemes: (1) Hard Negative Mixing (HNM), (2) Debiased Contrastive Learning (DCL), (3) Hardness-Biased Negative Mining (HBNM), and (4) Conditional Negative Mining (CNM).

## 2.2 Discussions on Representative GCL Methods

We give a brief summary of existing representative GCL methods as shown in Table 1 and discuss how these implementations fit into our proposed paradigm. It should be noted that negative mining strategies have received scant attention in current GCL literature and thus are omitted in the table.

**Dual branches vs. single branch.** We notice that most work leverages a dual-branch architecture following SimCLR [8] that augments the original graph twice to form two views and designates positive samples across two views. For some global-local CL methods like DGI [13] and GMI [15], they employ an architecture with only one branch. In this case, negative samples are obtained by corrupting the original graph. Different from the aforementioned *augmentation* schemes that generate congruent pairs to model *the joint distribution* of positive pairs, we resort to the term *corruption functions*, which approximate *the product of marginals*.

**Stronger augmentation.** Unlike GRACE [17] and GraphCL [16] that employ uniform edge/feature perturbation, GCA [18] proposes to perform adaptive augmentation based on importance scores of scores and feature dimensions. In this work, to involve less hyperparameters as possible, we focus on uniform transformation only.

**Variants of contrasting modes.** GMI [15] extends DGI [13] by further considering the agreement between raw node/edge features and node/edge representations. Because it requires much more computational resources, our experiments exclude this implementation of contrasting mode. In addition, there are several recent methods [27, 28] involve contrasting between local/global and *context* representations, which are usually derived from graph clustering algorithms. Considering the generality of the experiments, we shall leave it as a future direction.

**Contrastive architectures without explicit negative samples.** Recently, some GCL approaches relying on no negative samples have been proposed. BGRL [25] employs an asymmetric framework composing of online and offline encoders and directly use cosine similarity of two outputs as the self-supervision signal. GBT [26] proposes to construct a correlation matrix along the latent dimension instead of negative samples. In light of these methods, in this benchmarking study, we further conduct additional analysis on (1) the Bootstrapping Latent loss (BL), (2) Barlow Twins (BT) loss, and (3) VICReg loss.

## 3 Empirical Studies

The following section presents the empirical studies of graph contrastive learning. We comprehensively evaluate the model performance on different configurations. In the following section, we first introduce the experimental configurations and then summarize the results and observations regarding each particular component in the proposed paradigm. For details on the evaluational protocols and implementations, we refer readers of interest to Appendix B.

Table 2: Classification performance with different topology and feature augmentations. The best performance is highlighted in boldface and the second-to-best underlined. OOM indicates Out-Of-Memory on a 24GB GPU.

| | Aug. | Node | | | | Graph | | | |
|---|---|---|---|---|---|---|---|---|---|
| | | Wiki | CS | Physics | Computers | PTC_MR | PROTEINS | REDDIT-B | IMDB-M |
| | None | 71.73 ± 0.29 | 90.76 ± 0.05 | 93.69 ± 0.73 | 80.62 ± 0.62 | 50.22 ± 1.56 | 71.37 ± 0.48 | 68.64 ± 0.19 | 47.81 ± 0.14 |
| Topo. | EA | 79.42 ± 0.24 | 92.73 ± 0.10 | 94.77 ± 0.05 | 83.40 ± 0.64 | 58.34 ± 1.55 | 73.95 ± 0.90 | 79.40 ± 0.94 | 49.76 ± 0.64 |
| | ER | 78.61 ± 0.30 | 91.90 ± 0.06 | 95.21 ± 0.05 | **87.84 ± 0.76** | **60.24 ± 2.27** | 74.32 ± 0.42 | 83.30 ± 0.65 | 49.70 ± 0.48 |
| | EF | 75.70 ± 0.42 | 92.99 ± 0.15 | 94.88 ± 0.06 | 86.68 ± 0.73 | 58.22 ± 2.48 | 73.77 ± 0.42 | 83.69 ± 0.24 | 49.94 ± 0.21 |
| | ND | 79.57 ± 0.34 | 92.33 ± 0.12 | **95.99 ± 0.12** | 87.01 ± 0.54 | 59.77 ± 1.21 | **74.96 ± 0.90** | **84.76 ± 0.87** | **50.30 ± 0.51** |
| | PPR | 70.94 ± 0.43 | 92.25 ± 0.07 | OOM | 85.06 ± 0.53 | 56.08 ± 1.52 | 66.61 ± 0.90 | 70.66 ± 0.48 | 49.06 ± 0.37 |
| | MKD | 72.39 ± 0.49 | 92.62 ± 0.14 | OOM | 82.46 ± 0.58 | 56.66 ± 2.76 | 57.37 ± 0.34 | 71.87 ± 0.17 | 49.39 ± 0.25 |
| | RWS | **79.74 ± 0.34** | **93.48 ± 0.08** | 95.04 ± 0.11 | 87.60 ± 0.63 | 58.66 ± 2.39 | 72.13 ± 0.56 | 81.54 ± 0.78 | 49.96 ± 0.36 |
| Feat. | FM | 77.56 ± 0.46 | 91.55 ± 0.11 | 94.12 ± 0.21 | **85.05 ± 0.51** | 52.51 ± 2.43 | **73.50 ± 0.64** | 79.37 ± 0.03 | 49.47 ± 0.69 |
| | FD | **77.74 ± 0.34** | **91.83 ± 0.08** | 94.20 ± 0.16 | 84.93 ± 0.46 | **53.69 ± 2.36** | 72.95 ± 0.49 | **80.94 ± 0.12** | **49.82 ± 0.23** |

**Datasets.** We conduct experiments on a variety of datasets widely used in literature, ranging from academic networks to chemistry molecular datasets. For fair comparison, we closely follow previous studies on datasets preprocessing and experimental protocols [13, 14, 16–19, 25, 29–31].

**Evaluation configurations.** We mainly evaluate models with different design considerations on three benchmark tasks: (1) unsupervised node classification, (2) unsupervised graph classification, and (3) transfer learning. For all unsupervised tasks, we follow the linear evaluation scheme used by DGI [2] on nodes and InfoGraph [19] on graphs. The models are first trained in an unsupervised manner, and then the final embeddings are fed into a linear classifier to fit the labeled data. For transfer learning, we follow the settings in Hu et al. [31], where we first pretrain the encoder on part of the dataset and then finetune it on the rest. For all experiments, we run the model with ten random splits and report the averaged accuracies (%) as well as the standard deviation.

## 3.1 Data Augmentation

We first investigate how data augmentations affect the performance of GCL. Specifically, we apply different data augmentation functions to generate two views, leverage the InfoNCE objective, and contrast local-local representations. Except for augmentation functions used, all other settings are kept the same. The experimental results of employing different topology and feature augmentations are presented in Table 2. We further explore the use of compositional data augmentation schemes: (1) structure- and feature-level augmentations and (2) deterministic plus stochastic augmentations, where the results are summarized in Table 3.

**Obs. 1. Topology augmentations greatly affect model performance. Augmentation functions that produce sparser graphs generally lead to better performance.**

From Table 2, it is evident that the performance of GCL is highly dependent on the choice of topology augmentation functions. We observe that models that remove edges (ER, MDK, ND, PPR, and RWS), compared to models that add edges (EA), in general achieve better performance, which suggests that augmentation functions produce *sparser graph views* generally achieve better performance. We also find that RWS achieves better performance on node datasets, while ND favors graph tasks. We believe that by taking random walks we are able to better extract local structural patterns for one node. Since the graph datasets used in our study are of relatively small scales (< 500 nodes per graph), these random-walk-based sampling strategies may be confined and thus the simple node dropping (ND) scheme outperforms other augmentations on graph-level tasks.

To see how sparsity of the resulting views affect the performance, we further conduct sensitivity analysis on two models with ND, ER, and EA augmentations respectively, with varied dropping/adding probabilities on the CS dataset. The prediction accuracy along with the total number of edges in the produced graphs are plotted in Figure 2. From Figure 2a and 2b, we observe that model performance improves as more nodes are dropped and degenerates when the removal probability is too high. As seen in Figure 2c, the performance of EA augmentation downgrades greatly when more edges are added. In general, the result accords with our observations that many real-world graphs are inherently sparse [32, 33]. When too many edges are added, they connect nodes that are semantically unrelated, bringing noise to the generated graph view and thus deteriorate the quality of learned embeddings.

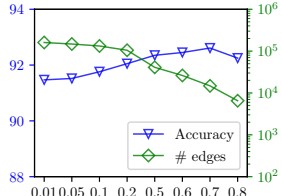
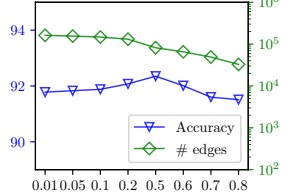
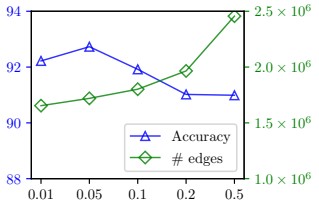

| (a) Node dropping probability | (b) Edge removing probability | (c) Edge adding probability |

Figure 2: Sensitivity analysis with varied topology augmentation probabilities on the CS dataset.

Table 3: Classification performance with different compositional augmentations.

(a) Joint augmentation on structure- and feature-level

| Aug. | Node | | | | Graph | | | |
|---|---|---|---|---|---|---|---|---|
| | Wiki | CS | Physics | Computers | PTC_MR | PROTEINS | REDDIT-B | IMDB-M |
| ER | 78.61 ± 0.30 | 91.90 ± 0.06 | 95.21 ± 0.05 | 87.84 ± 0.76 | 60.24 ± 2.27 | 74.32 ± 0.42 | 83.30 ± 0.65 | 49.70 ± 0.48 |
| +FM | 79.09 ± 0.41 | 92.45 ± 0.10 | 95.95 ± 0.12 | 88.15 ± 0.53 | **60.78 ± 1.52** | **75.47 ± 0.30** | 83.67 ± 0.40 | 50.17 ± 0.39 |
| +FD | **79.61 ± 0.38** | **93.63 ± 0.13** | **95.65 ± 0.06** | **88.42 ± 0.42** | 53.88 ± 2.30 | 75.03 ± 0.23 | **83.94 ± 0.60** | **50.30 ± 0.56** |
| ND | 79.57 ± 0.34 | 92.33 ± 0.12 | 95.99 ± 0.12 | 87.01 ± 0.54 | **59.77 ± 1.21** | 74.96 ± 0.90 | 84.76 ± 0.87 | **50.30 ± 0.51** |
| +FM | **79.80 ± 0.34** | 92.61 ± 0.14 | 96.01 ± 0.05 | 87.17 ± 0.58 | 58.23 ± 0.99 | **75.65 ± 0.27** | **84.94 ± 0.97** | 49.82 ± 0.59 |
| +FD | 79.68 ± 0.40 | **92.81 ± 0.07** | **96.15 ± 0.04** | **89.58 ± 0.40** | 57.28 ± 2.33 | 75.21 ± 0.88 | 84.64 ± 0.34 | 50.18 ± 0.50 |
| EA | 79.42 ± 0.24 | 92.73 ± 0.10 | 94.77 ± 0.05 | 83.40 ± 0.64 | 58.34 ± 1.55 | 73.95 ± 0.90 | 79.40 ± 0.94 | 49.76 ± 0.64 |
| +FM | 79.69 ± 0.45 | 93.47 ± 0.07 | **95.47 ± 0.05** | 83.81 ± 0.91 | **58.55 ± 1.99** | 74.04 ± 0.40 | 79.98 ± 2.29 | 50.47 ± 0.38 |
| +FD | **79.80 ± 0.17** | **93.62 ± 0.07** | 94.94 ± 0.12 | **84.63 ± 0.41** | 55.11 ± 1.41 | **74.89 ± 0.47** | **81.87 ± 1.86** | **50.55 ± 0.22** |
| RWS | 79.74 ± 0.34 | 93.48 ± 0.08 | 95.04 ± 0.11 | 87.60 ± 0.63 | 58.66 ± 2.39 | 72.13 ± 0.56 | 81.54 ± 0.78 | 49.96 ± 0.36 |
| +FM | **80.21 ± 0.30** | **94.35 ± 0.08** | **95.99 ± 0.08** | 87.63 ± 0.31 | **59.00 ± 1.88** | **74.88 ± 0.44** | 81.14 ± 1.27 | **50.74 ± 0.31** |
| +FD | 79.78 ± 0.17 | 93.98 ± 0.09 | 95.66 ± 0.06 | **89.38 ± 0.37** | 56.85 ± 1.05 | 73.28 ± 0.38 | **82.90 ± 1.33** | 50.39 ± 0.30 |

(b) Composition of stochastic and deterministic augmentations

| Aug. | Node | | | | Graph | | | |
|---|---|---|---|---|---|---|---|---|
| | Wiki | CS | Physics | Computers | PTC_MR | PROTEINS | REDDIT-B | IMDB-M |
| PPR | 70.94 ± 0.43 | 92.25 ± 0.07 | OOM | 85.06 ± 0.53 | 56.08 ± 1.52 | 66.61 ± 0.90 | 70.66 ± 0.48 | 49.06 ± 0.37 |
| +ER | 76.65 ± 0.42 | 92.41 ± 0.11 | — | 88.06 ± 0.62 | **57.65 ± 2.48** | **73.76 ± 0.38** | **73.81 ± 0.48** | 49.47 ± 0.24 |
| +EA | 77.71 ± 0.16 | 92.86 ± 0.16 | — | 86.29 ± 0.91 | 55.31 ± 1.80 | 68.55 ± 2.93 | 67.14 ± 4.82 | **50.47 ± 0.31** |
| +ND | **78.16 ± 0.51** | 92.77 ± 0.17 | — | **89.11 ± 0.33** | 56.19 ± 1.03 | 72.04 ± 1.96 | 72.40 ± 5.90 | 50.28 ± 0.27 |
| +FD | 74.12 ± 0.58 | **93.55 ± 0.09** | — | 86.49 ± 0.55 | 54.26 ± 1.71 | 72.51 ± 0.65 | 72.86 ± 0.30 | 49.28 ± 0.44 |
| MKD | 72.39 ± 0.49 | 92.62 ± 0.14 | OOM | 82.46 ± 0.58 | 56.66 ± 2.76 | 57.37 ± 0.34 | 71.87 ± 0.17 | 49.39 ± 0.25 |
| +ER | **78.42 ± 0.40** | 92.75 ± 0.05 | — | 89.60 ± 0.77 | **60.97 ± 2.25** | 71.51 ± 0.40 | 74.10 ± 0.45 | 49.63 ± 0.28 |
| +EA | 77.68 ± 0.55 | 92.71 ± 0.09 | — | 83.57 ± 1.08 | 54.60 ± 1.45 | 65.81 ± 3.01 | 65.65 ± 1.08 | **50.61 ± 0.17** |
| +ND | 77.35 ± 0.46 | 92.81 ± 0.05 | — | **89.67 ± 0.48** | 57.25 ± 1.87 | 72.40 ± 1.19 | **76.65 ± 2.91** | 50.41 ± 0.26 |
| +FD | 73.83 ± 0.42 | **93.83 ± 0.14** | — | 85.10 ± 0.92 | 54.46 ± 2.58 | **72.76 ± 0.79** | 73.82 ± 0.98 | 49.11 ± 0.68 |

**Obs. 2. Feature augmentations bring additional benefits to GCL when discriminatory features are available.**

From Table 2, we observe that the performance of models employing feature augmentations solely is inferior to that use topology augmentations. Also, we see that in general FD outperforms FM, which suggests the use of dropping features in an element-wise manner, though the performance increments are very limited.

Furthermore, we kindly note that graph datasets PTC_MR, RDT-B, and IMDB-M include no initial node features. Following previous work [19], we use 1-dimensional one's vector instead. It is seen that the performance gain of employing feature augmentations is not significant on these datasets that carry no informative node features. We thus posit that feature augmentation is helpful when discriminatory features are available. Along with Observation 1, we conclude that data augmentations in GCL should be adaptive to properties of the datasets, so the contrastive signal can better capture intrinsic patterns of graph structures and attributes, which corroborates the previous study [18].

**Obs. 3. Compositional augmentations at both structure and attribute level benefit GCL most. Deterministic augmentation schemes should be used along with stochastic augmentations.**

Table 4: Performance with different contrastive objectives and contrastive modes. L–L, G–L, and G–G denote local-local, global-local, and global-global contrasting modes. The best performing results for objectives (row-wise) and contrasting modes (column-wise) are highlighted in boldface and underline respectively.

(a) Node classification performance

| Obj. | Wiki | | CS | | Physics | | Computers | |
| --- | --- | --- | --- | --- | --- | --- | --- | --- |
| | L–L | G–L | L–L | G–L | L–L | G–L | L–L | G–L |
| InfoNCE | **79.09 ± 0.15** | 77.73 ± 0.94 | **92.45 ± 0.83** | 90.60 ± 0.06 | **95.95 ± 0.92** | 93.23 ± 0.96 | **88.15 ± 0.59** | 76.24 ± 0.93 |
| JSD | 78.83 ± 0.95 | **78.71 ± 0.19** | 92.18 ± 1.00 | **91.31 ± 0.62** | 94.32 ± 0.28 | **94.12 ± 0.04** | 82.02 ± 0.76 | **78.27 ± 0.05** |
| TM | 78.42 ± 0.88 | 76.53 ± 0.85 | 91.91 ± 0.31 | 90.11 ± 0.61 | 94.11 ± 0.60 | 92.78 ± 0.12 | 69.67 ± 0.88 | 76.38 ± 0.75 |

(b) Graph classification performance

| Obj. | PTC-MR | | | PROTEINS | | | REDDIT-B | | | IMDB-M | | |
| --- | --- | --- | --- | --- | --- | --- | --- | --- | --- | --- | --- | --- |
| | L–L | G–L | G–G | L–L | G–L | G–G | L–L | G–L | G–G | L–L | G–L | G–G |
| InfoNCE | **61.21 ± 1.92** | 60.21 ± 1.87 | 60.08 ± 1.14 | **75.01 ± 0.25** | 74.92 ± 0.53 | 75.31 ± 0.21 | 83.12 ± 0.59 | 82.62 ± 0.42 | **83.45 ± 0.23** | **50.22 ± 0.21** | 50.12 ± 0.25 | 49.23 ± 0.54 |
| JSD | 59.89 ± 2.39 | **60.39 ± 1.73** | 56.43 ± 2.51 | 74.11 ± 0.32 | 74.08 ± 0.40 | 74.32 ± 0.23 | **82.66 ± 0.73** | 81.98 ± 0.32 | 82.23 ± 0.47 | 50.02 ± 0.47 | 48.17 ± 0.28 | 49.34 ± 0.77 |
| TM | 54.92 ± 1.76 | 57.32 ± 2.31 | **61.53 ± 0.72** | 73.77 ± 0.47 | 71.31 ± 1.08 | 72.34 ± 1.16 | 79.32 ± 0.31 | 79.10 ± 0.83 | 83.21 ± 0.42 | 49.85 ± 0.69 | 48.40 ± 0.72 | **50.87 ± 0.13** |

From Table 3a, we observe that in most cases where meaningful node features are available, the use of feature augmentation in addition to structure augmentation benefits GCL, demonstrating that both topology and structures are important for learning graph representations. For datasets without informative node features, using feature augmentation may bring adverse effects, which corroborates our Observation 2. Also, we find that two feature augmentation schemes FM and FD perform similarly.

We also find that although two deterministic augmentation functions PPR and MDK do not perform well (cf. Table 2), joint utilization of stochastic and deterministic augmentation attains promising performance, echoing the design in MVGRL [14]. Recall that our contrastive objective is essentially aimed to discriminate between samples from the data distribution and samples from some "noise" distribution [34, 35]. Therefore, a stochastic augmentation scheme is needed to better approximate the noise distribution.

## 3.2 Contrasting Modes and Contrastive Objectives

The next experiments are concerned with how contrasting modes and contrastive objectives affect the model performance. We train the model with different contrasting modes and contrastive objectives, with topology augmentation set to ER and feature augmentation to FM. We also conduct experiments based on ND and FD, and we see no clear difference than that on ER and FM. Except that embedding sizes are fixed, other hyperparameters are tuned to obtain the best performance under each experiment for fair comparison. Table 4a and 4b present the experimental results on unsupervised classification tasks, and Table 5 presents performance on transfer learning (trained with the InfoNCE objective).

**Obs. 4. Downstream tasks of different granularities favor different contrasting modes.**

What stands out from the table is that contrasting local-local pairs achieves the best performance on node-level classification, while global-global and global-local modes perform better on graph-level tasks. Also, from Table 5 we observe similar trends on transfer learning of graph classification, where local-global and global-global contrasting modes generally perform better. This suggests us to choose a contrasting mode corresponding to granularity of the end task, i.e. local-local for node-level tasks and global-local or global-global for graph-level tasks.

For graph-level tasks, contrasting the global-local embedding pairs explicitly encodes global information into global representations. However, its performance on node-level tasks is inferior to that uses

Table 5: Downstream performance on transfer learning of graph classification.

| Mode | BBBP | Tox21 | ToxCast | SIDER | ClinTox | MUV | HIV | BACE | PPI |
| --- | --- | --- | --- | --- | --- | --- | --- | --- | --- |
| L–L | 72.26 ± 0.81 | 74.79 ± 0.52 | 62.33 ± 0.53 | 60.43 ± 1.35 | 75.11 ± 2.16 | 71.41 ± 1.97 | 77.96 ± 0.72 | **78.53 ± 1.84** | 65.53 ± 0.58 |
| G–L | **73.12 ± 0.62** | **74.89 ± 0.43** | **62.45 ± 0.67** | **60.62 ± 1.08** | 75.01 ± 2.37 | **71.62 ± 2.02** | 78.02 ± 0.84 | 77.43 ± 1.28 | 66.46 ± 0.98 |
| G–G | 71.52 ± 0.57 | 73.32 ± 0.78 | 62.34 ± 0.66 | 60.40 ± 0.99 | **76.03 ± 2.33** | 70.24 ± 1.93 | **78.24 ± 0.99** | 76.23 ± 1.48 | **67.23 ± 0.65** |

local-local mode. We may explain the results from the perspective of the optimization objective. The global-local mode is essentially a proxy for local-local CL, provided that the readout function $r$ is expressive enough [36, 13]. However, in reality, the injectivity property of the readout function is hard to fulfill [37]. Therefore, the readout function may not distill enough information from node-level embeddings, leading to performance gap between node- and graph-level tasks.

In accordance with the presented results, recent studies [38, 39] have made a similar finding for learning visual representations. They demonstrate that being trained on instance-level pretext tasks (i.e. contrasting image-level embeddings in the same batch), current CL models have suboptimal performance in fine-grained tasks, e.g., semantic segment that requires pixel-level details.

**Obs. 5. The use of InfoNCE objective leads to consistent improvements across all settings yet requires a rather large number of negative samples.**

From Table 4a, it is seen that the InfoNCE achieve the best performance among contrastive objectives, which has already been shown effective by many recent methods [4, 7, 8, 40, 41].

Among the studied three objectives, recent studies have already revealed that the InfoNCE loss has an intrinsic ability to perform hard negative samples [23]. Particularly, one very recent study in computer vision [42] shows that the use of a temperature parameter $\tau$ in the InfoNCE objective acts as an adjustment factor to exert penalties on hard negative samples. To verify this on GCL, we further conduct sensitivity analysis on this temperature parameter on both node (CS) and graph (IMDB-M) datasets as shown in Figure 3. We observe that with the increase of $\tau$, the performance improves at first and downgrades later, with not much performance fluctuation. According to

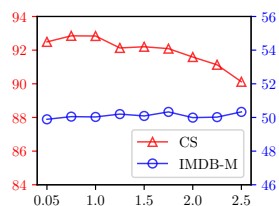

Figure 3: Sensitivity analysis of $\tau$ in the InfoNCE objective.

Wang and Liu [42], the InfoNCE objective pays less attention to hard negatives as $\tau$ increases. This hardness-aware behavior demonstrates the importance of *striking a balance* between separation of hardest negative samples ($\tau \to 0^+$) and global uniformity ($\tau \to \infty$) on GCL.

We also note that we obtain close-to-optimal performance on several graph datasets when we train the model using the triplet loss to contrast global-global embeddings. This could be explained by the fact that the triplet loss is known to be sensitive to hard negatives [43]. In the global-global mode, the contrasting pairs are graph embeddings in the batch, which are more distinct (harder) than each other, compared to node pairs. Therefore, we suspect that it provides more informative signals to hard negative samples that boosts model learning.

### 3.3 Negative Mining Strategies

Following the previous section, we probe the explicit use of negative mining strategies on top of contrastive objectives, which essentially measure the hardness of each negative pair and upweight hard negative samples. We train four models on three node classification datasets (Wiki, CS, and Computers) using the local-local mode with the InfoNCE objective. The results are presented in Figure 4.

**Obs. 6. Existing negative mining techniques bring limited benefit to GCL.**

From the figure, we see that the mixup strategy (HNM) consistently improves the performance on all three datasets, though the improvements are marginal. With more hard negatives being synthesized, the contrastive objective gives larger weights to harder negatives (i.e. embeddings close to the anchor), which improves the discriminative power of the contrastive model. The other three strategies, while slight improvement can be observed under certain hyperparameter configurations, do not in general improve the InfoNCE baseline by much.

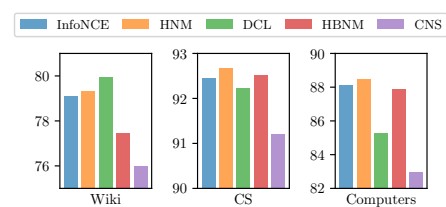

Figure 4: Performance comparison with different negative mining strategies.

In existing formulation of these four negative mining techniques, the sample hardness is measured by inner product of sample embeddings. Since we are in a fully unsupervised setting, no label (class) information can be accessed. Under existing contrasting modes, for one anchor sample, the

contrastive objective pushes all different representations away, *irrespective of their real semantic relations*. What is even worse, most GNN models tend to produce similar embeddings for neighboring nodes regardless of semantic classes [44–47], which may further bias the selection of hard negatives. Therefore, we argue that there is disagreement between semantic similarity and example hardness. By selecting hard negative samples merely according to similarity measure of embeddings, these hard negatives are potentially *positive samples*, which produces adverse learning signals to the contrastive objective. We kindly find that our discovery reminiscent to one very recent study in visual contrastive learning [42], which recommends an adaptive scheduling scheme for the temperature parameter when using InfoNCE as the contrastive objective, so that hard but false negatives could be tolerated as the training progresses.

## 3.4 Summary of Additional Experiments

We conduct additional experiments regarding large-scale evaluation and negative-sample-free contrastive objectives. Due to space limitation, we summarize our key findings here and readers of interest may refer to Appendix C for details.

**Large-scale evaluation.** Current CL models suffer from large computational burden due to the need of negative samples. Since the experiments are limited to small- to medium-scale datasets, we conduct a further study assessing how existing GCL work scales to large-scale graphs using two open graph benchmark [48] datasets. From the experimental results, we find that our main conclusions are still applicable to large-scale datasets.

**Experiments on negative-sample-free contrastive objectives** Besides contrastive objectives that rely on negative samples, we experiment with three negative-sample-free objectives: Bootstrapping Latent (BL) loss, Barlow Twins (BT) loss, and VICReg loss. We observe from the results that these three objectives greatly reduce the computational burden since no explicit negative samples are constructed. More surprisingly, Barlow Twins and VICReg losses are even able to achieve better performance compared to their negative-sample-based counterparts.

## 3.5 Discussions on Limitations and Future Directions

Due to limited space, some limitations of our work need to be acknowledged.

- **Limited design considerations.** In this work, we consider limited design considerations, namely four design dimensions. An issue that is not addressed in this study is what role do many other model-specific factors, e.g., whether to employ a projection head in the InfoNCE objective and what graph encoders should be employed, play in GCL.
- **Limited downstream tasks.** Our empirical study only includes experiments on node- and graph-level classification and graph-level transfer learning; a boarder range of end tasks of different granularities, e.g., link prediction and community detection, may be more beneficial to draw convincing conclusions.
- **Lack of theoretical justification.** Our work only presents empirical studies which has thrown up many questions in need of further theoretical justification for better understanding the mechanisms underlying GCL, for instance, how to appropriately measure and select hard negative samples for contrastive objectives in the graph domain.

Our empirical findings also suggest several future directions that may be helpful for improving GCL.

- **Towards automated augmentation.** We understand that topology augmentation is of paramount importance to GCL. However, existing work leverages manually designed ad-hoc augmentation strategies, which may result in suboptimal performance. Recent studies in graph structure learning establish a principled way to learn optimal structures of graph-structured data [33], which we argue could be used for automatically learn augmentation functions suitable for GCL tasks.
- **Understanding the performance gap between pretext and downstream tasks.** We empirically demonstrate the correlation between the choice of end tasks and contrastive objectives, yet calls for a thorough understanding for the performance gap between pretext and downstream tasks. We have found that there is some progress in this regard [49], but it is far from fully explored.
- **Structure-aware negative sampling.** Unlike in computer vision fields, similar visual features may naturally correlate to closer semantic categories, measuring the hardness through embedding

similarities in graph-structured data is more difficult. A series of earlier work in network embedding proposes solutions from structural aspects [50–52]. However, how to integrate rich structure information for modeling better negative distributions for GCL is still left unexplored.

## 4    Conclusion

In this paper, we first present a taxonomy for GCL, where we categorize existing work to four aspects: data augmentations, contrasting modes, contrastive objective, and negative sampling techniques. Then, we analyze the choices in the design space for each aspect by extensively study the empirical performance of models under different design choices over a comprehensive set of benchmarking tasks and datasets. Our rigorous analysis results in interesting findings about the interplay of design dimensions in GCL. We also provide a handy toolbox PyGCL to facilitate the implementation and experimenting of GCL models. We hope our empirical study provides several practical guidelines for future research in this vigorous field.

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
