# OpenReview forum: "An Empirical Study of Graph Contrastive Learning"
_NeurIPS.cc/2021/Track/Datasets_and_Benchmarks/Round1 — Submitted to NeurIPS 2021 Datasets and Benchmarks Track (Round 1)_

### Official Review · Reviewer_LNr1 · 2021-07-03
**A benchmarking study that has a couple of questionable statements**

**Rating:** 5
**Confidence:** 4
**Correctness:** 1. In section 3.1, observation 1, it …

**Strengths:**

1. As a work in the scope of the graph learning community, this paper provides a unified description of the GCL framework. This can help foster future studies on GCL and help them communicate their contributions in the same terms.

2. Within this description, the authors present their findings as suggestions for specific graph datasets and learning tasks. Specifically, they outline the most significant component in terms of the effectiveness of the algorithm, how each variant affects the final performance, which design choices work better for various types of data or tasks.

3. Such studies typically help researchers facilitate comparisons and find directions to improve and invent approaches.

**Weaknesses:**

1. Possible design choices and learning tasks. The authors study only a limited number of methods for each stage, and most are relatively basic. Also, experiments only include graph and node classification tasks. This

2. Large-scale datasets evaluation. The graph-level datasets used in the study include up to 2000 graphs; node-level datasets are of small or medium scale. Although these datasets are standard benchmarks for graph learning papers, experiments on less known graph datasets would benefit the practical value of this paper.

3. Some of the conclusions drawn from the experimental results are questionable, see Correctness paragraph.

**Additional Feedback:**

Major questions and feedback:
1. The choice of metrics. Why was accuracy chosen as the primary metric for the results? Is accuracy a good fit for the considered tasks? Would the results change for a different metric?

2. Section 3.2: "We train the model with different contrasting modes and contrastive objectives, with topology augmentation set to ER and feature augmentation to FM." Why this choice of augmentations if this combination is not the best performing? Primarily this concerns the choice of FM (Feature Masking) vs. FD (Feature Dropout).

Suggestions and comments:
1. Possibly, some of my concerns could be eliminated with box plots showing the confidence intervals.

2. The text should be revised to spot several typos.

Additional info from the authors does address some of the concerns, however at this point this study should be revised and structured more carefully and resubmitted.

**Clarity:**

The paper has several typos and bits hard to follow, but the overall text quality is satisfactory.

**Documentation:**

The authors included the code with necessary implementations for each set of experiments. A separate python package was developed to ease future research in this direction.

**Ethics:**

No ethical concerns are spotted.

**Relation To Prior Work:**

The related work section is somewhat short and uninformative. For a reader outside of the graph learning community, it can be challenging to assess the impact and completeness of this paper. With a lack of context, it is hard to follow the description of design choices.

**Summary And Contributions:**

The paper describes a study of the graph contrasting learning (GCL) approaches. The authors propose decomposing the GCL framework into separate stages: data augmentation, contrasting modes, contrastive objectives, and negative sampling strategies. This study sets apart these stages from the graph model used as a backbone for GCL inference.

The paper includes a benchmarking of each stage, presenting results on multiple graph datasets for both node-level and graph-level tasks.

As a list of contributions, the paper mentions several bits of advice on using specific algorithms at each GCL stage and provides a general overview of use-cases at each stage.

---

### Official Review · Reviewer_Fc8B · 2021-07-05
**Valuable contribution; needs better documentation**

**Rating:** 6
**Confidence:** 2
**Correctness:** Yes, the experiment design and evalua…
**Clarity:** Yes, the paper is well-written, excep…

**Strengths:**

The paper clearly categorizes recent approaches and studies the effect of different factors using a well-laid out and extensive experiments on several datasets. This is a commendable empirical study that seems quite valuable to the community. Accompanying code makes it easy to build over these conclusions.

**Weaknesses:**

Better descriptions in both the paper (supplemental pdf seems to be missing) and the library are needed. For example Section 3:Datasets actually does not cite or explain the studied datasets in any way (what is "Wiki", "CS", "Physics" in Table 2?). It is also not clear how generalizable these conclusions are across different model architectures, or what models were actually employed in the evaluation. E.g. comparing train_graph_BGRL.py train_graph_BGRL_GRACE.py in the code reveals nearly identical code with an extra BatchNorm layer. Is this significant in any way?

**Additional Feedback:**

- Supplemental material is simply a copy of the main paper. Is this a mistake?
- The code URL is not accessible. Is the code going to become available upon publication?
- The library requires *significantly better* documentation and README. Which commands correspond to which tables, as well as general code overview, e.g. enough to make it easy to replicate experiments on new datasets with different underlying models. To become a common tool, some refactoring seems in order.  E.g. train_graph_BGLR.py and train_graph_BGLR_GRACE.py are near duplicates of each other, which is a bad software engineering practice.

**Documentation:**

Not applicable.

**Ethics:**

No.

**Relation To Prior Work:**

Current work clearly categorizes previous approaches and presents extensive analysis. No new models are proposed.

**Summary And Contributions:**

The paper presents a taxonomy of approaches in recent research on GLC and systematically studies the value of different components, including data augmentation, contrasting modes, objectives and negative sampling strategies on several common datasets. A library able to reproduce the experiments will be made available.

---

### Official Review · Reviewer_sB1y · 2021-07-05
**An Empirical Study of Graph Contrastive Learning**

**Rating:** 6
**Confidence:** 3
**Clarity:** The paper is well-written.

**Strengths:**

The paper provides a unified description of the GCL framework, which helps foster future studies on GCL. The paper presents their findings for specific graph datasets and learning tasks. The paper is well written  and easy to follow

**Weaknesses:**

Some details are missed in the paper. It is not clear that which models were actually employed in the evaluation. More discussions and analysises should be given in the evaluation processes.

**Additional Feedback:**

The code URL is not accessible.

The paper should be revised for grammar errors.

**Correctness:**

Missing details of Observation 3 and Table 3.  It is doubtful of the claim that ND gains significant improvement from feature augmentations.

**Documentation:**

The authors included the code with necessary implementations for each set of experiments.

**Relation To Prior Work:**

The paper categorizes previous approaches.

**Summary And Contributions:**

The paper first identify several critical design considerations within a general GCL paradigm, including augmentation functions, contrasting modes, contrastive objectives, and negative sampling techniques. And to understand the interplay of different GCL components, the paper conducts extensive, controlled experiments over a set of benchmark tasks on datasets across various domains.

---

### Decision · Program_Chairs · 2021-07-26

**Decision:**

Reject

**Comment:**

In this paper the authors perform an empirical study of graph contrastive learning (GCL) to understand how the myriad of design decisions that contribute to various GCL designs effect performance.  Reviewers generally found the unified framework valuable and the results interesting. That said, there were relatively consistent questions about the datasets and model architectures chosen, and the implications these have on the generality the results.  While the results are interesting, as an empirical evaluation of the algorithms this seems like a better fit for the main conference track.